# Burden of Illness beyond Mortality and Heart Failure Hospitalizations in Patients Newly Diagnosed with Heart Failure in Spain According to Ejection Fraction

**DOI:** 10.3390/jcm12062410

**Published:** 2023-03-21

**Authors:** Carlos Escobar, Beatriz Palacios, Victoria Gonzalez, Martín Gutiérrez, Mai Duong, Hungta Chen, Nahila Justo, Javier Cid-Ruzafa, Ignacio Hernández, Phillip R. Hunt, Juan F. Delgado

**Affiliations:** 1Cardiology Department, University Hospital La Paz, 28046 Madrid, Spain; 2AstraZeneca Farmaceutica, 28050 Madrid, Spain; 3Evidera, London W6 8BJ, UK; 4AstraZeneca, Gaithersburg, MD 20878, USA; 5Evidera, 113 21 Stockholm, Sweden; 6Karolinska Institute, Department of Neurobiology, Care Sciences, and Society, 171 77 Stockholm, Sweden; 7Evidera, 08005 Barcelona, Spain; 8Atrys Health, 28002 Madrid, Spain; 9Cardiology Department, University Hospital 12 de Octubre, CIBERCV, 28041 Madrid, Spain

**Keywords:** dapagliflozin, heart failure, MACE, myocardial infarction, SGLT2 inhibitors, stroke

## Abstract

Objective: The objective of this study was to describe the rates of adverse clinical outcomes, including all-cause mortality, heart failure (HF) hospitalization, myocardial infarction, and stroke, in patients newly diagnosed with HF to provide a comprehensive picture of HF burden. Methods: This was a retrospective and observational study, using the BIG-PAC database in Spain. Adults, newly diagnosed with HF between January 2013 and September 2019 with ≥1 HF-free year of enrolment prior to HF diagnosis, were included. Results: A total of 19,961 patients were newly diagnosed with HF (43.5% with reduced ejection fraction (EF), 26.3% with preserved EF, 5.1% with mildly reduced EF, and 25.1% with unknown EF). The mean age was 69.7 ± 19.0 years; 53.8% were men; and 41.0% and 41.5% of patients were in the New York Heart Association functional classes II and III, respectively. The baseline HF treatments included beta-blockers (70.1%), renin–angiotensin system inhibitors (56.3%), mineralocorticoid receptor antagonists (11.8%), and SGLT2 inhibitors (8.9%). The post-index incidence rates of all-cause mortality, HF hospitalization, and both combined were 102.2 (95% CI 99.9–104.5), 123.1 (95% CI 120.5–125.7), and 182 (95% CI 178.9–185.1) per 1000 person-years, respectively. The rates of myocardial infarction and stroke were lower (26.2 [95% CI 25.1–27.4] and 19.8 [95% CI 18.8–20.8] per 1000 person-years, respectively). Conclusions: In Spain, patients newly diagnosed with HF have a high risk of clinical outcomes. Specifically, the rates of all-cause mortality and HF hospitalization are high and substantially greater than the rates of myocardial infarction and stroke. Given the burden of adverse outcomes, these should be considered targets in the comprehensive management of HF. There is much room for improving the proportion of patients receiving disease-modifying therapies.

## 1. Introduction

Heart failure (HF) is associated with a high risk of mortality and frequent hospital admissions [1]. In fact, around 20% of patients with a recent diagnosis of HF are expected to die during the following year [2]. HF hospitalizations are a common complication of patients with HF. HF hospitalizations represent an inflection point in the evolution of HF, as vulnerability to complications is particularly high during the first months after hospitalization for HF [3,4].

However, patients with HF are not only at risk of cardiovascular death and HF hospitalization but also other adverse outcomes, such as major adverse cardiovascular events (MACEs), myocardial infarction, and stroke [4]. Thus, HF remains a frequent complication of patients with prior myocardial infarction [5], and it is also common in the overall population with coronary artery disease [6]. In fact, coronary artery disease is one of the main risk factors for HF development [7]. Of note, chronic HF is a risk factor for the development of ischemic stroke by itself, beyond its association with atrial fibrillation [8,9]. In this context, the early initiation of guideline-directed HF therapy is crucial to decrease the comprehensive HF burden, including not only the risks of HF hospitalizations and death but also the risks of other adverse events [5,6,7,8,9]. However, only a few HF clinical trials have analyzed the effects of active treatments on endpoints other than HF hospitalizations and cardiovascular death, including MACEs, myocardial infarction, and stroke [10].

However, it is important to ascertain whether there are differences in the clinical profile, management, and risk of events (including MACEs, myocardial infarction, and stroke), not only in the whole HF population but also stratified by HF EF phenotypes (i.e., HF with a reduced ejection fraction (HFrEF), HF with preserved EF (HFpEF), and HF with mildly reduced EF (HFmrEF)). However, this has not been well studied, and more information is warranted [11,12,13].

The objective of this study was to describe the clinical characteristics of the population and the incidence rates of HF hospitalization and MACEs (including myocardial infarction, stroke, and all-cause mortality) in an overall incident HF cohort of newly diagnosed patients, who were also stratified by EF subgroup. In addition, the factors associated with increasing the risk of HF hospitalization and death, as well as the evolution of HF treatment over time, were determined.

## 2. Methods

This was a retrospective observational cohort study that analyzed the information provided in the BIG-PAC database in Spain. This database collects information from integrated and computerized medical records of a total of 1.8 million patients in 7 Spanish Autonomous Communities. Data are available for the year 2012 onwards and are updated monthly. Many studies have demonstrated the validity and applicability of this database [13,14,15]. This study was approved by the Investigation Ethics Committee of HM Hospitals in Madrid, Spain. No informed consent was required, as this study was a secondary data study using fully anonymized data.

Patients aged 18 years or older, with at least 1 year of enrollment in the database prior to the index date, were included. HF was defined as having at least 1 new HF diagnosis (ICD-9/ICD-10 codes) in the inpatient (any position) or outpatient records between 1 January 2013 and 30 September 2019. The index date was the date of the first HF diagnosis. Patients with chronic stage V kidney disease who required dialysis at any time before the index date were excluded from the study. Patients with HF were classified into different phenotypes according to left ventricular EF: HFpEF was defined as an EF value of ≥50% (subtype 1: EF 50 to < 60%; subtype 2: EF ≥ 60%), HFrEF was defined as an EF value of ≤40%, HFmrEF was defined as an EF value > 40% and < 50%, and HF with an unknown EF (HFuEF) included patients without echocardiograph data.

At baseline, biodemographic data, cardiovascular risk factors, vascular disease, and other comorbidities, as well as newly prescribed HF treatments, were recorded. The baseline clinical characteristics, treatments, and outcomes were analyzed in the overall HF population and according to EF phenotype. If a patient was prescribed 2 different drug classes on the same date or took a combination of pills, they were included in both treatment classes. The evolution of HF treatment within the first year after the index date was determined. The primary outcome was a composite of HF hospitalization and all-cause mortality and its individual components. HF hospitalizations were determined as an inpatient admission (primary or any diagnosis) with an ICD-10 code for any HF. Furthermore, the occurrences of 2 additional adverse clinical outcomes, i.e., myocardial infarction, stroke, all-cause mortality, and its composite (MACEs) were also analyzed from 1 to 7 years from the index date.

### Statistical Analysis

HF incidence (per 1000 person-years) was calculated by dividing the number of patients who received a new HF diagnosis during the study period by the total person-time contributed by all adult patients in the database without prevalent HF. The incidence rates of clinical outcomes were calculated in the overall cohort of patients with HF and according to EF phenotype. HF incidence was calculated in the overall cohort of patients with HF and according to EF phenotype.

The baseline characteristics of the patients were summarized using descriptive statistics. For continuous variables, the number of patients, mean, and standard deviation were reported. Frequency distributions with quantity and percentages were reported for categorical variables. To analyze the relationship between the continuous variables amongst the EF phenotypes, a 2-sample t-test was used for variables normally distributed, and the Mann–Whitney U test was used for those non-normally distributed. The chi-square test was used for categorical variables. Wald contrast was used for the incident rates and event rates. The McNemar test was used to compare HF treatment changes over time. A statistical significance level of 0.05 was applied.

The incidence rates of clinical outcomes were calculated as the total number of new events of interest divided by the total person-time at risk. Patients were followed from the index date until the earliest occurrence of the event of interest, death, loss to follow-up, or study end date. The incidence rates were reported per 1000 person-years over the entire follow-up duration and by year after the index date. The incidence rates and cumulative incidence rates for the primary endpoint of the composite of HF hospitalization and all-cause mortality, as well as for the composite of MACE outcomes (myocardial infarction, stroke, and all-cause mortality) and individual components, were calculated for the overall HF cohort and stratified by EF phenotype. Kaplan–Meier plots were constructed to illustrate the time to the first event (in days). The recurrent event rates (per 1000 person-years) were calculated for myocardial infarction, stroke, and HF hospitalizations. A minimum of 30 days was required between outcome events of the same type for them to be considered separate events.

A multivariable Cox proportional hazards regression model was used to assess the associations between the baseline clinical characteristics and treatments and the risk of the composite endpoint of hospitalization for HF and all-cause mortality in the overall HF cohort. The regression analyses consisted of a 2-step assessment: in the first step, univariable Cox models were constructed for each baseline covariate, and unadjusted hazard ratios and the corresponding 95% confidence intervals were calculated for each covariate. Covariates with a univariate *p*-value of <0.1 were included in the multivariate models. After all covariates were identified, they were combined in a multivariable regression model using stepwise backwards selection to exclude covariates that became nonsignificant. 

All data were analyzed using the statistical package SPSS v25.0 (SPSS Inc., Chicago, IL, USA).

## 3. Results

The incidence of HF over the study period was 3.2 per 1000 person-years (1.4 for HFrEF, 0.9 for HFpEF, 0.2 for HFmrEF, and 0.8 per 1000 person-years for HFuEF), increasing from 2.7 per 1000 person-years in 2013 to 3.7 per 1000 person-years in 2018. 

The baseline characteristics and treatments in the incident 2013–2019 HF cohort are presented in Table 1. Of the 19,961 patients, 43.5% had HFrEF, 26.3% had HFpEF, and 5.1% had HFmrEF; in the remaining 25.1%, EF was unknown. Overall, the mean age was 69.7 ± 19.0 years; 53.8% were men; and 41.0% and 41.5% were in the New York Heart Association (NYHA) functional classes II and III, respectively. The most common comorbidities were hypertension (59.1%), coronary artery disease (33.1%), atrial fibrillation (28.2%), type 2 diabetes (27.6%), and chronic kidney disease (26.7%). The baseline HF treatments included beta-blockers (70.1%), renin–angiotensin system inhibitors (56.3%), mineralocorticoid receptor antagonists (11.8%), and sodium–glucose cotransporter-2 inhibitors (SGLT2is) (8.9%). There were relevant differences in the clinical profiles and HF management according to HF phenotype. Compared to the patients with HFrEF, the patients with HFpEF were older (73.4 ± 18.6 vs. 65.6 ± 18.6 years; *p* = 0.001), more predominantly female (66.2% vs. 34.1%; *p* < 0.001) and had a higher prevalence of atrial fibrillation (35.5% vs. 23.5%; *p* < 0.001) at baseline. The patients with HFrEF had a higher prevalence of type 2 diabetes (25.8% vs. 28.3%; *p* < 0.001), chronic kidney disease (22.5% vs. 30.8%; *p* < 0.001), coronary artery disease (26.2% vs. 38.7%; *p* < 0.001), stroke (6.0% vs. 12.3%; *p* < 0.001), and peripheral artery disease (3.2% vs. 5.1%; *p* < 0.001) at baseline. The prevalence of comorbidities in the patients with HFmrEF was intermediate, falling between that of the patients with HFrEF and that of the patients with HFpEF. Among the patients with HFpEF those with EF 50% to < 60% had higher rates of diabetes (38.3% vs. 24.0%; *p* < 0.001), prior myocardial infarction (17.8% vs. 11.0%; *p* < 0.001), stroke (8.1% vs. 4.9%; *p* < 0.001), and peripheral artery disease (4.6% vs. 2.5%; *p* < 0.001), compared to those with EF ≥ 60%. Similarly, prescriptions of disease-modifying HF drugs (i.e., beta-blockers, renin–angiotensin system inhibitors, mineralocorticoid receptor antagonists, and SGLT2is) were more frequent in patients with HFrEF than in those with the other HF EF phenotypes. All HF drugs were significantly more frequently prescribed in the patients surviving to the 12-month followup regardless of HF phenotype. However, all HF drugs were more commonly taken among patients with HFrEF during the whole followup period (Table 2 and Appendix A).

In all patients with HF, the incidence rates of all-cause mortality, HF hospitalization, and the combined endpoint of HF hospitalization and all-cause mortality were 102.2 (95% CI 99.9–104.5), 123.1 (95% CI 120.5–125.7), and 182 (95% CI 178.9–185.1) per 1000 person-years, respectively. The rates of myocardial infarction, stroke, and MACEs were 26.2 (95% CI 25.1–27.4), 19.8 (95% CI 18.8–20.8), and 148.9 (95% CI 145.9–151.9) per 1000 person-years, respectively. Although events were common in all HF EF phenotypes, the incidence rates were higher in the patients with HFrEF. Among the patients with HFpEF, although the rates of HF hospitalization and the composite outcome of HF hospitalization and mortality was independent of EF, the rates of myocardial infarction, stroke, all-cause mortality, and MACEs were more common in the patients with EF 50–60% compared to those with EF ≥ 60% (Table 3, Figure 1, and Appendix A). The evolution of adverse outcomes over the 7-year follow-up from the index date is shown in Appendix A.

The factors associated with the risk of the composite outcome hospitalization of HF and all-cause mortality in the overall HF cohort are shown in Figure 2. HFrEF (vs. HFmrEF or HFpEF), an increased age, the male sex, prior myocardial infarction, unstable angina, percutaneous or surgical revascularization, stroke, type 2 diabetes, COPD, and the use of digoxin were associated with a higher risk of the composite outcome.

## 4. Discussion

This study shows that, in Spain, patients with HF are of older age (nearly 60% of all patients with HF in this study were older than 65 years), particularly those with HFpEF, and that they have many comorbidities. The rates of HF hospitalization, all-cause mortality, and their composite were particularly high. Although lower, the rates of myocardial infarction and stroke were also significant. Although the prescription of HF drugs has improved in recent years, many patients were not taking the disease-modifying HF drugs by the end of the follow-up.

Previous studies have shown that the current incidence of HF in Europe is around 3–5/1000 person-years [1,4,16]. In our study, we found an incidence of 3.2 per 1000 person-years for all HF EF phenotypes, in line with previous data, with the annual incidence increasing over the study period. The number of patients with prevalent HF is predicted to increase in the coming years because of population aging and patients with HF surviving longer as a result of improved treatment options [17,18].

Our study included nearly 20,000 patients newly diagnosed with HF. Among those patients with known EF, 58.1% had HFrEF, 35.1% had HFpEF, and 6.8% had HFmrEF. Although there were relevant disparities in the clinical profiles according to EF phenotype (i.e., the patients with HFpEF and HFmrEF were older and more commonly women than the patients with HFrEF, as expected), overall, the patients with HF were old and had many comorbidities. The relative proportions of the EF phenotype vary considerably across studies, as do the clinical profiles. This may be related to the clinical setting that patients attend, as well as reflecting geographic differences. Thus, patients of cardiologists are usually younger and have higher rates of HFrEF than patients of internal medicine departments, who are usually older and more likely to have HFpEF [11,12,13,14,19]. The BIG-PAC database collects information from integrated and computerized medical records from both inpatient and primary care [13,14,15]. Our data provide a more balanced and comprehensive picture of the Spanish population newly diagnosed with HF.

The primary outcomes of our study, all-cause mortality and HF hospitalization, were designed to be similar to the outcomes of recent clinical trials [10,20,21,22,23,24]. The rates of all-cause mortality and HF hospitalization in this real-world population (10.2 and 12.3 per 100 patient-years, respectively) are higher than those reported in recent trials [10,20,21,22,23,24]. In a pooled analysis of the DAPA-HF and DELIVER trials, in the placebo arm, the rates of all-cause mortality and total HF hospitalizations per 100 patient-years were 8.3 and 11.4, respectively [10]. The higher rates in our study may be due not only to the longer follow-up time but also to the complexity of patients included in real-life studies being higher than that of patients included in clinical trials [25,26].

With regard to HF treatments, European guidelines recommend the use of renin–angiotensin system inhibitors (preferably sacubitril/valsartan), beta-blockers, SGLT2is, and mineralocorticoid receptor antagonists as first-line therapies for patients with HFrEF (and with a lower strength of evidence for patients with HFmrEF). By contrast, for patients with HFpEF, diuretics for treating congestive symptoms are the only drug class specifically recommended by European guidelines, along with the adequate management of comorbidities [4]. However, since the publication of these guidelines, two clinical trials have demonstrated that some SGLT2is can also reduce the risks of cardiovascular death and HF hospitalization in patients with HFpEF [23,24]. In addition, a recent pooled meta-analysis of the DAPA-HF and DELIVER trials has shown that the SGLT2i dapagliflozin significantly decreases the risks of cardiovascular death by 14%, all-cause death by 10%, and total HF hospitalizations by 29% in the whole spectrum of patients with HF, regardless of left ventricular EF [10].

Our study shows that, across the EF spectrum, at baseline, only 70% of patients were taking beta-blockers, 55% were taking renin–angiotensin system inhibitors, 12% were taking mineralocorticoid receptor antagonists, and 9% were taking SGLT2is (our study period was prior to the first approval of SGLT2is for HF in the European Union). In addition, the increase in the proportion of patients taking HF drugs after the one-year follow-up was modest (2%, 9%, 7%, and 1%, respectively), and, even 12 months after diagnosis, many patients remained untreated with guideline-directed therapies. Furthermore, a recent study has shown that the initiation of novel guideline-directed medical therapies, such as dapagliflozin or sacubitril/valsartan, is delayed compared with other that of HF drugs and that only a small proportion of patients receive the target doses of HF drugs requiring uptitration [27]. However, the use of some drugs (such as digoxin) that have not demonstrated benefits for mortality are still widely used [28]. Therefore, to decrease the risks of death and HF hospitalization, more efforts are necessary to reduce the gaps between recommended therapies impacting on morbidity and mortality and clinical practice [13].

Importantly, our study also analyzed the risks of other cardiovascular outcomes: the incidence rates of myocardial infarction, stroke, and their composite MACE with all-cause mortality were 26, 20 and 149 per 1000 person-years, respectively, with the rates of MACE increasing over time. This information is relevant, as it has not been well studied in previous studies. For example, it is well-known that myocardial infarction is an important cause of HF, particularly HFrEF, and when acute coronary syndrome is complicated with HF, the risk of recurrent adverse cardiovascular events markedly increases [5], yet coronary artery disease goes unrecognized in a great number of patients with HF (either HFrEF or HFpEF) [29,30]. In addition, our study found in a multivariate analysis that a history of myocardial infarction or coronary revascularization was associated with a higher risk of death or HF hospitalization. In this context, comprehensive management targeting both HF and coronary artery disease is mandatory [6,31,32]. Of note, a recent clinical trial has shown that, compared with optimal medical therapy, percutaneous revascularization does not provide additional benefits to patients with severe ischemic left ventricular systolic dysfunction, indicating that the best approach in these patients is the early implementation of disease-modifying HF therapies [4,33].

A recent meta-analysis has shown that patients with HF have a relative risk of developing ischemic stroke of 2.3 vs. those without HF. About 4% of prevalent patients have prior ischemic stroke, and the annual incidence is 2.16% (vs. 3.56 per 100 persons—3.64 per 100 person-years—within the first year of follow-up in our study) [9]. This higher risk of stroke markedly increases with the coexistence of atrial fibrillation [4]. Of note, ischemic stroke significantly increases morbidity and mortality in HF, independently of the presence of atrial fibrillation [34]. As a result, to reduce the risk of stroke in patients with HF, it is not only important to implement treatment with HF drugs but to also implement treatment with anticoagulation drugs when required (i.e., in the presence of atrial fibrillation) [34,35]. 

Finally, the risk of MACEs was high in our study, and, consequently, it should be considered a target by itself. As a result, when treating patients with HF, one should consider not only reducing the risks of cardiovascular death and HF hospitalization but also the risk of MACEs. Of note, the rates of MACEs were higher in the patients with HFrEF than in the patients with HFpEF, with the patients with HFmrEF having intermediate values; furthermore, in the HFpEF subgroup, the rates of MACEs were higher in the patients with EF 50–60% than in the patients with EF ≥ 60%. In this context, the pooled analysis of the DAPA-HF and DELIVER trials showed that the addition of dapagliflozin to standard therapy in patients with HF was associated with a significant reduction in MACE risk of 10% [10].

## 5. Study Limitations

This study has some limitations. As this was a retrospective study, only data that were collected in the electronic clinical history of patients could be recorded, leading to the underdiagnosis of some variables. However, the high number of patients could have reduced this potential bias. Furthermore, the results of this study can only be extended to patients with similar clinical profiles and healthcare management.

## 6. Conclusions

Patients with HF are of an older age, have many comorbidities, and have a high risk of adverse outcomes, including not only death and HF hospitalization but also myocardial infarction, stroke, and particularly MACE. For this reason, all these adverse events should be considered targets in the comprehensive management of patients as soon as HF is diagnosed. In this context, there is much room for improving the proportion of patients receiving disease-modifying HF drugs, which would translate into a robust reduction in HF burden.

## Figures and Tables

**Figure 1 jcm-12-02410-f001:**
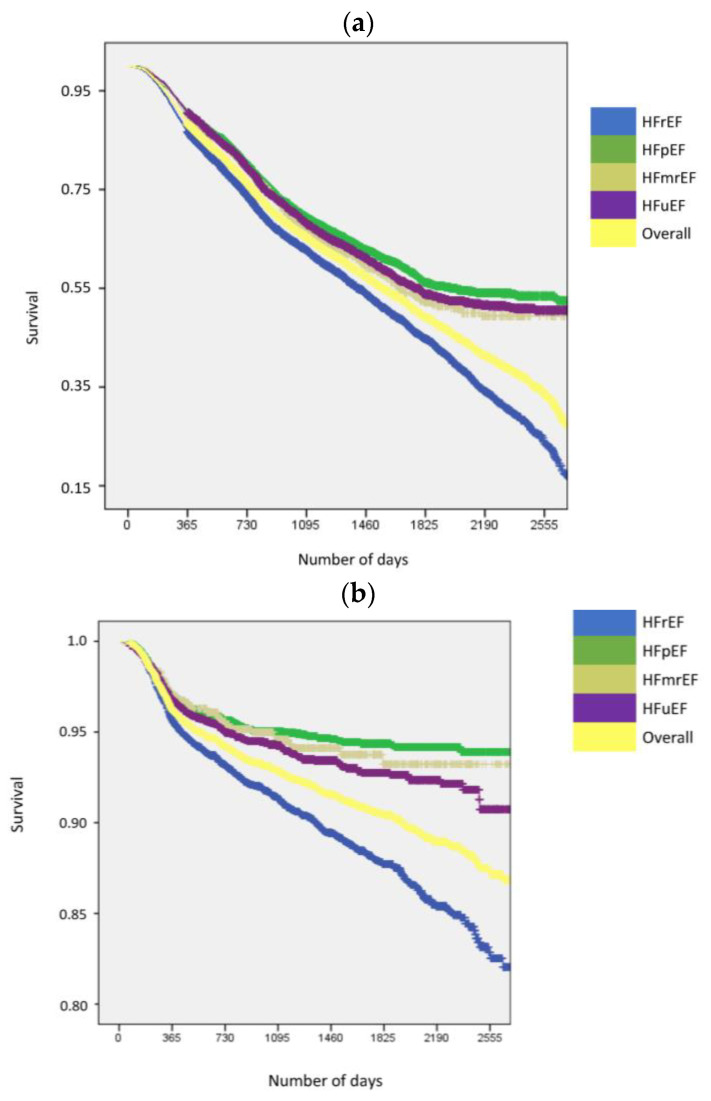
Survival free of adverse clinical outcomes. (**a**) Survival free of the composite MACE outcome: overall and by EF subgroup. (**b**) Survival free of stroke: overall and by EF subgroup. (**c**) Survival free of all-cause mortality: overall and by EF subgroup. (**d**) Survival free of myocardial infarction: overall and by EF subgroup. (**e**) Survival free of HF hospitalization: overall and by EF subgroup. (**f**) Survival free of HF hospitalization and all-cause mortality: overall and by EF subgroup. Abbreviations: EF = ejection fraction; HF = heart failure; HFmrEF = heart failure with a mildly reduced ejection fraction; HFpEF = heart Failure with a preserved ejection fraction; HFrEF = heart failure with a reduced ejection fraction; HFuEF = heart failure with an unknown ejection fraction; MACE = major adverse cardiovascular event.

**Figure 2 jcm-12-02410-f002:**
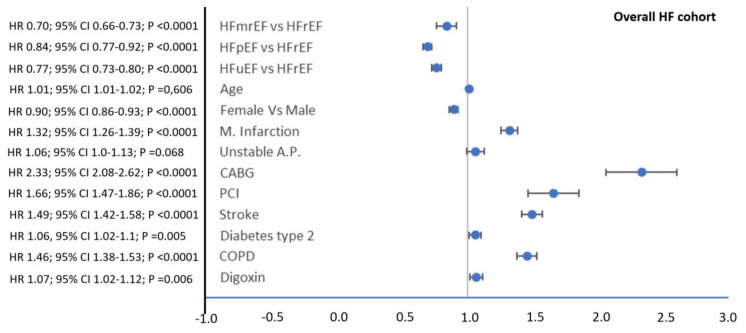
Cox regression analysis for associations between specific patient characteristics and the risk of the composite outcome of HF hospitalization and all-cause mortality (time to first event, either HF hospitalization or all-cause death) in the overall HF cohort. Abbreviations: CABG = coronary artery bypass graft; COPD = chronic obstructive pulmonary disease; HF = heart failure; HFmrEF = heart failure with a mildly reduced ejection fraction; HFpEF = heart failure with a preserved ejection fraction; HFrEF = heart failure with a reduced ejection fraction; HFuEF = heart failure with an unknown ejection fraction; M. Infarction = myocardial infarction; PCI = percutaneous intervention; Unstable A.P. = unstable angina pectoris.

**Table 1 jcm-12-02410-t001:** Baseline clinical characteristics and treatments in the incident HF cohort (index date 2013–2019).

	HF Incident Cohort (n = 19,961; 100%)	HFrEF (n = 8678; 43.5%)	HFmrEF (n = 1022; 5.1%)	HFpEF (n = 5244; 26.3%)	HFpEF (50 to <60%) (n = 1833; 9.2%)	HFpEF (≥60%) (n = 3411; 17.1%)	HFuEF (n = 5017; 25.1%)	*p*-Value (HFmrEF vs. HFrEF)	*p*-Value (HFpEF vs. HFrEF)	*p*-Value (EF ≥ 60% vs. 50–60%)
**Biodemographic data**										
Age. Years, mean ± SD	69.7 ± 19.0	65.61 ± 18.6	72.3 ± 18.8	73.4 ± 18.6	73.2 ± 18.4	73.5 ± 18.7	72.3 ± 18.9	<0.001	0.001	0.300
<45 years, n (%)	2653 (13.3)	1550 (17.9)	113 (11.1)	485 (9.3)	156 (8.5)	329 (9.6)	505 (10.1)	<0.001	<0.001	0.131
45–64 years, n (%)	5713 (28.6)	2718 (31.3)	262 (25.6)	1350 (25.7)	492 (26.8)	858 (25.2)	1383 (27.6)	<0.001	<0.001	0.370
65–74 years, n (%)	2829 (14.2)	1314 (15.1)	148 (14.5)	691 (13.2)	252 (13.7)	439 (12.9)	676 (13.5)	0.453	0.002	0.161
75–84 years, n (%)	2953 (14.8)	1386 (16.0)	145 (14.2)	716 (13.7)	255 (13.9)	461 (13.5)	706 (14.1)	0.587	<0.001	0.805
≥85 years, n (%)	5813 (29.1)	1710 (19.7)	354 (34.6)	2002 (38.2)	678 (37.0)	1324 (38.8)	1747 (34.8)	<0.001	<0.001	0.460
Sex (male), n (%)	10,731 (53.8)	5719 (65.9)	433 (42.4)	1772 (33.8)	608 (33.2)	1164 (34.1)	2807 (56.0)	<0.001	<0.001	0.486
NYHA functional classification								<0.001	<0.001	0.558
Class I, n (%)	2669 (13.4)	1158 (13.3)	129 (12.6)	670 (12.8)	238 (13.0)	432 (12.7)	712 (14.2)	0.559	0.337	0.741
Class II, n (%)	8182 (41.0)	3040 (35.0)	456 (44.6)	2704 (51.6)	940 (51.3)	1764 (51.7)	1982 (39.5)	<0.001	<0.001	0.765
Class III, n (%)	8274 (41.5)	4032 (46.5)	384 (37.6)	1749 (33.4)	619 (33.8)	1130 (33.1)	2109 (42.0)	<0.001	<0.001	0.638
Class IV, n (%)	551 (2.8)	326 (3.8)	36 (3.5)	71 (1.4)	23 (1.3)	48 (1.4)	118 (2.4)	0.793	<0.001	0.649
Unknown, n (%)	285 (1.4)	122 (1.4)	17 (1.7)	50 (1.0)	13 (0.7)	37 (1.1)	96 (1.9)	0.487	0.019	0.182
**Cardiovascular risk factors**										
Hypertension, n (%)	11,793 (59.1)	5293 (61.0)	626 (61.3)	2947 (56.2)	1055 (57.6)	1892 (55.5)	2927 (58.3)	0.630	<0.001	0.069
Dyslipidemia, n (%)	8959 (44.9)	3813 (43.9)	459 (44.9)	2308 (44.0)	829 (45.2)	1479 (43.4)	2379 (47.4)	0.947	0.167	0.203
Diabetes type 1, n (%)	741 (3.7)	356 (4.1)	45 (4.4)	169 (3.2)	94 (5.1)	75 (2.2)	171 (3.4)	0.598	<0.001	<0.001
Diabetes type 2, n (%)	5511 (27.6)	2459 (28.3)	277 (27.1)	1354 (25.8)	609 (33.2)	745 (21.8)	1421 (28.3)	0.178	<0.001	<0.001
**Vascular disease**										
Coronary artery disease, n (%)	6602 (33.1)	3361 (38.7)	318 (31.1)	1376 (26.2)	475 (25.9)	901 (26.4)	1547 (30.8)	<0.001	<0.001	0.814
Myocardial Infarction, n (%)	3002 (15.0)	1387 (16.0)	123 (12.0)	702 (13.4)	326 (17.8)	376 (11.0)	790 (15.8)	<0.001	<0.001	<0.001
Unstable angina, n (%)	1667 (8.4)	872 (10.1)	77 (7.5)	305 (5.8)	99 (5.4)	206 (6.0)	413 (8.2)	0.002	<0.001	0.232
PCI, n (%)	369 (1.9)	160 (1.8)	16 (1.6)	108 (2.1)	37 (2.0)	71 (2.1)	85 (1.7)	0.356	0.697	0.860
CABG, n (%)	331 (1.7)	124 (1.4)	16 (1.6)	101 (1.9)	34 (1.9)	67 (2.0)	90 (1.8)	0.797	0.082	0.670
Chronic kidney disease by stage, n (%)	5337 (26.7)	2674 (30.8)	310 (30.3)	1181 (22.5)	418 (22.8)	763 (22.4)	1172 (23.4)	0.438	<0.001	0.893
Stage Unknown, n (%)	2221 (11.1)	1126 (13.0)	128 (12.5)	485 (9.3)	171 (9.3)	314 (9.2)	482 (9.6)			
Stage I, n (%)	122 (0.6)	60 (0.7)	5 (0.5)	30 (0.6)	15 (0.8)	15 (0.4)	27 (0.5)			
Stage II, n (%)	561 (2.8)	271 (3.1)	30 (2.9)	144 (2.8)	45 (2.5)	99 (2.9)	116 (2.3)			
Stage III, n (%)	1830 (9.2)	908 (10.5)	116 (11.4)	404 (7.7)	150 (8.2)	254 (7.4)	402 (8.0)			
Stage IV, n (%)	437 (2.2)	214 (2.5)	18 (1.8)	90 (1.7)	33 (1.8)	57 (1.7)	115 (2.3)			
Stage V, n (%)	166 (1.0)	95 (1.1)	13 (1.1)	28 (0.9)	4 (0.9)	24 (0.9)	30 (0.6)			
Stroke, n (%)	2014 (10.1)	1069 (12.3)	95 (9.3)	315 (6.0)	149 (8.1)	166 (4.9)	535 (10.7)	0.010	<0.001	<0.001
Peripheral arterial disease, n (%)	943 (4.7)	441 (5.1)	31 (3.0)	168 (3.2)	84 (4.6)	84 (2.5)	303 (6.0)	0.002	<0.001	<0.001
**Other comorbidities**										
COPD, n (%)	2634 (13.2)	1278 (14.7)	97 (9.5)	626 (11.9)	205 (11.2)	421 (12.3)	633 (12.6)	<0.001	<0.001	0.145
Atrial fibrillation, n (%)	5637 (28.2)	2043 (23.5)	304 (29.8)	1861 (35.5)	658 (35.9)	1203 (35.3)	1429 (28.5)	<0.001	<0.001	0.758
Cancer, n (%)	2377 (11.9)	1077 (12.4)	117 (11.5)	567 (10.8)	190 (10.4)	377 (11.1)	616 (12.3)	<0.001	<0.001	0.357
Hepatic disease, n (%)	960 (4.8)	471 (5.4)	37 (3.6)	261 (5.0)	85 (4.6)	176 (5.2)	191 (3.8)	0.007	0.073	0.332
**Medications**										
**HF drugs**										
Diuretics, n (%)	13,845 (69.4)	6175 (71.2)	632 (61.8)	3542 (67.5)	1258 (68.6)	2284 (67.0)	3496 (69.7)	<0.001	<0.001	0.666
Beta-blockers, n (%)	13,992 (70.1)	6257 (72.1)	707 (69.2)	3414 (65.1)	1206 (65.8)	2208 (64.7)	3614 (72.0)	0.028	<0.001	0.579
ACEi/ARB, n (%)	10,026 (50.2)	4967 (57.2)	370 (36.2)	1856 (35.4)	636 (34.7)	1220 (35.8)	2833 (56.5)	<0.001	<0.001	0.418
ARNI, n (%)	1219 (6.1)	605 (7.0)	49 (4.8)	215 (4.1)	96 (5.2)	119 (3.5)	350 (7.0)	<0.001	<0.001	0.410
MRA, n (%)	2360 (11.8)	1068 (12.3)	127 (12.4)	531 (10.1)	188 (10.3)	343 (10.1)	634 (126)	<0.001	<0.001	0.356
Digoxin, n (%)	4007 (20.1)	1960 (22.6)	172 (16.8)	910 (17.4)	330 (18.0)	580 (17.0)	965 (19.2)	<0.001	<0.001	0.057
Ivabradine, n (%)	1218 (6.1)	616 (7.1)	37 (3.6)	249 (4.8)	106 (5.8)	143 (4.2)	316 (6.3)	<0.001	<0.001	0.930
Hydralazine and nitrate, n (%)	14 (0.07)	7 (0.08)	1 (0.10)	4 (0.08)	3 (0.2)	1 (0.0)	2 (0.04)	0.955	0.477	0.953
**Other cardiovascular drugs**										
Lipid-lowering drugs, n (%)	10,664 (53.4)	4992 (57.5)	588 (57.5)	2554 (48.7)	864 (47.1)	1690 (49.5)	2530 (50.4)	0.066	<0.001	0.901
Antiplatelets, n (%)	7319 (36.7)	3612 (41.6)	359 (35.1)	1582 (30.2)	538 (29.4)	1044 (30.6)	1766 (35.2)	<0.001	<0.001	0.851
Anticoagulants, n (%)	5337 (26.7)	1868 (21.5)	240 (23.5)	1814 (34.6)	616 (33.6)	1198 (35.1)	1415 (28.2)	0.001	<0.001	0.769
Calcium channel blockers	3349 (16.8)	822 (9.5)	149 (14.6)	1449 (27.6)	515 (28.1)	934 (27.4)	929 (18.5)	<0.001	<0.001	0.493
Nitrates, n (%)	2419 (12.1)	1065 (12.3)	144 (14.1)	570 (10.9)	211 (11.5)	359 (10.5)	640 (12.8)	0.003	0.018	0.051
Nicorandil, n (%)	17 (0.09)	4 (0.05)	0	7 (0.13)	2 (0.1)	5 (0.1)	6 (0.1)	0.346	0.533	0.953
**Antihyperglycemic drugs**										
Metformin, n (%)	4069 (20.4)	1881 (21.7)	190 (18.6)	1009 (19.2)	340 (18.5)	669 (19.6)	989 (19.7)	0.078	0.005	0.634
DPP4i, n (%)	2442 (12.2)	1219 (14.1)	82 (8.0)	470 (9.0)	158 (8.6)	312 (9.1)	671 (13.4)	<0.001	<0.001	0.757
Sulfonylurea, n (%)	2342 (11.7)	1052 (12.1)	149 (14.6)	566 (10.8)	188 (10.3)	378 (11.1)	575 (11.5	0.041	0.120	0.256
SGLT2is, n (%)	1779 (8.9)	729 (8.4)	86 (8.4)	505 (9.6)	190 (10.4)	315 (9.2)	459 (9.2)	0.003	0.463	0.836
Insulin, n (%)	1462 (7.3)	640 (7.4)	71 (7.0)	369 (7.0)	128 (7.0)	241 (7.1)	382 (7.6)	0.412	0.102	0.083
GLP1-RA, n (%)	240 (1.2)	101 (1.2)	19 (1.9)	72 (1.4)	31 (1.7)	41 (1.2)	48 (1.0)	0.315	0.149	0.795
Other glucose-lowering drugs, n (%)	864 (4.3)	443 (5.1)	29 (2.8)	160 (3.1)	57 (3.1)	103 (3.0)	232 (4.6)	0.033	<0.001	0.017
**Individual active prescriptions**										
0, n (%)	8 (0.04)	1 (0.01)	0	6 (0.1)	4 (0.2)	2 (0.1)	1 (0.02)	<0.001	<0.001	<0.001
1, n (%)	81 (0.4)	23 (0.27)	2 (0.2)	37 (0.7)	12 (0.7)	25 (0.7)	19 (0.4)			
2, n (%)	376 (1.9)	91 (1.1)	44 (4.3)	155 (3.0)	38 (2.1)	117 (3.4)	86 (1.7)			
3, n (%)	1227 (6.2)	324 (3.7)	110 (10.8)	475 (9.1)	149 (8.1)	326 (9.6)	318 (6.3)			
4, n (%)	2543 (12.7)	860 (9.9)	186 (18.2)	880 (16.8)	326 (17.8)	554 (16.2)	617 (12.3)			
≥5, n (%)	15,726 (78.8)	7379 (85.0)	680 (66.5)	3691 (70.4)	1304 (71.2)	2387 (69.9)	3976 (79.3)			

Patients with CKD stage V were excluded from the incident HF cohort. All treatments were assessed within 12 months before the index. Patients taking a combination of drugs were counted in each respective treatment class. Abbreviations: ACEi = angiotensin-converting enzyme inhibitors; ARB = angiotensin receptor II blockers; ARNI: angiotensin II receptor antagonist and a neprilysin inhibitor; CABG = coronary artery bypass graft; COPD = chronic obstructive pulmonary disease; DPP4i = dipeptidyl peptidase 4 inhibitors; GLP1-RA = glucagon-like peptide-1 receptor agonists; HF = heart failure; HFmrEF = heart failure with a mildly reduced ejection fraction; HFpEF = heart failure with a preserved ejection fraction; HFrEF = heart failure with a reduced ejection fraction; HFuEF = heart failure with an unknown ejection fraction; MRA: mineralocorticoid receptor antagonists; NYHA: New York Heart Association; PCI = percutaneous intervention; SGLT2is = sodium–glucose cotransporter-2 inhibitors. The continuous variable (age) was compared using a 2-sample t-test (variable normally distributed), and the chi-square test was performed to compare categorical variables.

**Table 2 jcm-12-02410-t002:** Evolution of HF treatment in the incident HF cohort (index date 2013–2019).

	HF Incident Cohort	HFrEF	HFmrEF	HFpEF	HFpEF (50 to <60%)	HFpEF (≥60%)	HFuEF	*p*-Value (HFmrEF vs. HFrEF)	*p*-Value (HFpEF vs. HFrEF)	*p*-Value (EF ≥ 60% vs. 50–60%)
**Baseline (n = 19,961)**										
Diuretics, n (%)	13,845 (69.4)	6175 (71.2)	632 (61.8)	3542 (67.5)	1258 (68.6)	2284 (67.0)	3496 (69.7)	<0.001	<0.001	0.666
Beta-blockers, n (%)	13,992 (70.1)	6257 (72.1)	707 (69.2)	3414 (65.1)	1206 (65.8)	2208 (64.7)	3614 (72.0)	0.028	<0.001	0.579
ACEi/ARB, n (%)	10,026 (50.2)	4967 (57.2)	370 (36.2)	1856 (35.4)	636 (34.7)	1220 (35.8)	2833 (56.5)	<0.001	<0.001	0.418
ARNI, n (%)	1219 (6.1)	605 (7.0)	49 (4.8)	215 (4.1)	96 (5.2)	119 (3.5)	350 (7.0)	<0.001	<0.001	0.410
MRA, n (%)	2360 (11.8)	1068 (12.3)	127 (12.4)	531 (10.1)	188 (10.3)	343 (10.1)	634 (126)	<0.001	<0.001	0.356
SGLT2is, n (%)	1779 (8.9)	729 (8.4)	86 (8.4)	505 (9.6)	190 (10.4)	315 (9.2)	459 (9.2)	<0.001	<0.001	0.836
Digoxin, n (%)	4007 (20.1)	1960 (22.6)	172 (16.8)	910 (17.4)	330 (18.0)	580 (17.0)	965 (19.2)	<0.001	<0.001	0.057
Ivabradine, n (%)	1218 (6.1)	616 (7.1)	37 (3.6)	249 (4.8)	106 (5.8)	143 (4.2)	316 (6.3)	0.955	0.477	0.930
Hydralazine and nitrate, n (%)	14 (0.07)	7 (0.08)	1 (0.10)	4 (0.08)	3 (0.2)	1 (0.0)	2 (0.04)	<0.001	<0.001	0.953
**6 months (n = 19,818)**										
Diuretics, n (%)	14,196 (71.6)	6303 (73.2)	651 (64.1)	3642 (70.0)	1279 (70.2)	2363 (69.9)	3600 (72.2)	<0.001	<0.001	0.929
Beta-blockers, n (%)	14,105 (71.2)	6289 (73.0)	717 (70.6)	3453 (66.4)	1219 (66.9)	2234 (66.1)	3646 (73.1)	0.118	<0.001	0.941
ACEi/ARB, n (%)	10,329 (52.1)	5069 (58.8)	386 (38.0)	1969 (37.9%)	681 (37.4)	1288 (38.1)	2905 (58.3)	<0.001	<0.001	0.175
ARNI, n (%)	1720 (8.7)	811 (9.4)	74 (7.3)	365 (7.0)	147 (8.1)	218 (6.4)	470 (9.4)	0.027	<0.001	0.956
MRA, n (%)	3035 (15.3)	1350 (15.7)	169 (16.6)	700 (13.5)	257 (14.1)	443 (13.1)	816 (16.4)	0.415	<0.001	0.616
SGLT2is, n (%)	1852 (9.4)	754 (8.8)	91 (9.0)	530 (10.2)	196 (10.8)	334 (9.9)	477 (9.6)	0.817	0.005	0.928
Digoxin, n (%)	4299 (21.7)	2097 (24.3)	189 (18.6)	982 (18.9)	350 (19.2)	632 (18.7)	1031 (20.7)	<0.001	<0.001	0.055
Ivabradine, n (%)	1346 (6.8)	670 (7.7)	43 (4.2)	278 (5.3)	118 (6.5)	160 (4.7)	355 (7.1)	<0.001	<0.001	0.387
Hydralazine and nitrate, n (%)	82 (0.4)	37 (0.4)	5 (0.5)	19 (0.4)	10 (0.5)	9 (0.3)	21 (0.4)	0.772	0.563	0.117
**12 months (n = 19,309)**										
Diuretics, n (%)	14,249 (73.8)	6320 (75.4)	661 (66.8)	3664 (72.3)	1283 (72.4)	2381 (72.3)	3604 (74.0)	<0.001	<0.001	0.929
Beta-blockers, n (%)	13,965 (72.3)	6215 (74.2)	712 (71.9)	3433 (67.8)	1209 (68.2)	2224 (67.5)	3605 (74.0)	0.192	<0.001	0.941
ACEi/ARB, n (%)	10,455 (54.2)	5069 (60.5)	405 (40.9)	2044 (40.3)	702 (39.6)	1342 (40.7)	2937 (60.3)	<0.001	<0.001	0.175
ARNI, n (%)	2156 (11.2)	1003 (12.0)	97 (9.8)	478 (9.4)	185 (10.4)	293 (8.9)	578 (11.9)	0.049	<0.001	0.637
MRA, n (%)	3655 (18.9)	1598 (19.1)	206 (20.8)	881 (17.4)	314 (17.7)	567 (17.2)	970 (19.9)	0.176	0.016	0.616
SGLT2i, n (%)	1895 (9.8)	783 (9.3)	95 (9.6)	526 (10.4)	189 (10.7)	337 (10.2)	491 (10.1)	0.774	0.048	0.865
Digoxin, n (%)	4481 (23.2)	2167 (25.9)	204 (20.6)	1025 (20.2)	368 (20.8)	657 (19.9)	1085 (22.3)	<0.001	<0.001	0.055
Ivabradine, n (%)	1471 (7.6)	719 (8.6)	48 (4.9)	306 (6.0)	125 (7.1)	181 (5.5)	398 (8.2)	<0.001	<0.001	0.233
Hydralazine and nitrate, n (%)	135 (0.7)	62 (0.7)	8 (0.8)	30 (0.6)	13 (0.7)	17 (0.5)	35 (0.7)	0.807	0.315	0.367

Abbreviations: ACEi = angiotensin-converting enzyme inhibitors; ARB = angiotensin receptor II blockers; ARNI: angiotensin II receptor antagonist and a neprilysin inhibitor; HF = heart failure; HFmrEF = heart failure with a mildly reduced ejection fraction; HFpEF = heart failure with a preserved ejection fraction; HFrEF = heart failure with a reduced ejection fraction; HFuEF = heart failure with an unknown ejection fraction; MRA: mineralocorticoid receptor antagonists; SGLT2is = sodium–glucose cotransporter-2 inhibitors. The chi-square test was performed to compare categorical variables.

**Table 3 jcm-12-02410-t003:** Rates of adverse clinical outcomes in the incident HF cohort.

	All HF (N = 19,961)	HFrEF (N = 8678)	HFmrEF (N = 1022)	HFpEF (N = 5244)	HFpEF (50 to <60%) (N = 1833)	HFpEF (≥60%) (N = 3411)	HFuEF (N = 5017)	*p*-Value (HFmrEF vs. HFrEF)	*p*-Value (HFpEF vs. HFrEF)	*p*-Value (EF ≥ 60% vs. 50–60%)
**Incidence Rates ^1^**										
**Myocardial infarction**								0.001	0.000	0.032
Patients with outcome (n)	1851	1027	85	372	149	223	367			
Total person-years	70,635	30,639	3630	18,951	6578	12,373	17,415			
Rate per 1000 person-years (95% CI)	26.2 (25.1–27.4)	33.5 (31.6–35.6)	23.4 (19–28.9)	19.6 (17.7–21.7)	22.7 (19.3–26.5)	18 (15.8–20.5)	21.1 (19–23.3)			
**Stroke**								<0.001	0.000	0.025
Patients with outcome (n)	1414	816	52	253	105	148	293			
Total person-years	71,553	31,224	3631	19,129	6642	12,487	17,569			
Rate per 1000 person-years (95% CI)	19.8 (18.8–20.8)	26.1 (24.4–28)	14.3 (10.9–18.7)	13.2 (11.7–14.9)	15.8 (13.1–19.1)	11.9 (10.1–13.9)	16.7 (14.9–18.7)			
**All-cause mortality**								<0.001	0.000	0.042
Patients with outcome (n)	6775	3605	295	1455	540	915	1420			
Total person-years	66,280	29,052	3324	17,546	6111	11,435	16,358			
Rate per 1000 person-years (95% CI)	102.2 (99.9–104.5)	124.1 (120.3–127.9)	88.7 (79.6–98.9)	82.9 (78.9–87.1)	88.4 (81.5–95.7)	80 (75.2–85.1)	86.8 (82.6–91.2)			
**Composite MACE**								0.000	0.000	0.006
Patients with outcome (n)	8136	4282	365	1755	658	1097	1734			
Total person-years	54,644	25,660	3026	16,167	5563	10,604	15,009			
Rate per 1000 person-years (95% CI)	148.9 (145.9–151.9)	166.9 (162.4–171.5)	120.6 (109.5–132.7)	108.6 (103.9–113.4)	118.3 (110–127)	103.5 (97.8–109.4)	115.5 (110.5–120.7)			
**HF hospitalization**								0.008	0.000	0.157
Patients with outcome (n)	7464	3757	399	1606	585	1021	1702			
Total person-years	60.651	25.968	3.106	16.530	5.720	10.810	15.047			
Rate per 1000 person-years (95% CI)	123.1 (120.5–125.7)	144.7 (140.5–149)	128.4 (117.1–140.7)	97.2 (92.7–101.8)	102.3 (94.7–110.4)	94.5 (89.1–100.1)	113.1 (108.2–118.3)			
**HF hospitalization and mortality ***								<0.001	0.000	0.127
Patients with outcome (n)	11,037	5536	551	2472	897	1575	2478			
Total person-years	60.651	25.968	3.106	16.530	5.720	10.810	15.047			
Rate per 1000 person-years (95% CI)	182 (178.9–185.1)	213.2 (208.2–218.2)	177.4 (164.3–191.2)	149.5 (144.2–155.1)	156.8 (147.6–166.5)	145.7 (139.2–152.5)	164.7 (158.8–170.7)			
**Event Rates ^2^**										
**Myocardial infarction**								0.003	0.070	0.072
Patients with outcome (n)	2873	1542	138	635	244	391	558			
Total person-years	71,433	31,082	3661	19,152	6652	12,500	17,538			
Rate per 1000 person-years (95% CI)	40.2 (38.8–41.7)	49.6 (47.3–52.1)	37.7 (32–44.4)	33.2 (30.7–35.8)	36.7 (32.4–41.5)	31.3 (28.4–34.5)	31.8 (29.3–34.5)			
**Stroke**								0.014	0.000	0.100
Patients with outcome (n)	2277	1245	97	459	175	284	476			
Total person-years	72,418	31,699	3653	19,331	6718	12,613	17,735			
Rate per 1000 person-years (95% CI)	31.4 (30.2–32.7)	39.3 (37.2–41.5)	26.6 (21.8–32.3)	23.7 (21.7–26)	26 (22.5–30.1)	22.5 (20.1–25.3)	26.8 (24.6–29.3)			
**HF: hospitalization**								0.000	<0.001	0.116
Patients with outcome (n)	13,567	7733	610	2658	967	1691	2566			
Total person-years	66.280	29.052	3.324	17.546	6.111	11.435	16.358			
Rate per 1000 person-years (95% CI)	204.7 (201.6–207.8)	266.2 (261.1–271.3)	183.5 (170.7–197)	151.5 (146.3–156.9)	158.2 (149.3–167.6)	147.9 (141.5–154.5)	156.9 (151.4–162.5)			

* After discharge. Abbreviations: CI = confidence interval; HF = heart failure; HFmrEF = heart failure with a mildly reduced ejection fraction; HFpEF = heart failure with a preserved ejection fraction; HFrEF = heart failure with a reduced ejection fraction; HFuEF = heart failure with an unknown ejection fraction; HR: hazard ratio; MACE = major adverse cardiovascular event; 95% CI = 95% confidence interval. ^1^ Incidence rates were defined as the total number of incident events of interest divided by the total person-time at risk; ^2^ Event rates were defined as the total number of events, including recurrent events, divided by the total person-time of follow-up. Wald contrast was used for rates.

## Data Availability

This was a secondary data study using the BIG-PAC^®^ database, and the data can be obtained upon reasonable request.

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
