# Peer review of "Burden of Illness beyond Mortality and Heart Failure Hospitalizations in Patients Newly Diagnosed with Heart Failure in Spain According to Ejection Fraction"

_jcm, 2023, doi:10.3390/jcm12062410_

Round 1
Reviewer 1 Report
jcm-2209374, Burden of Illness Beyond Mortality and Heart Failure Hospitalizations in Newly Diagnosed Heart Failure Patients in Spain According to Ejection Fraction by Carlos Escobar et al. The authors aimed in their study to describe patient characteristics, rates of adverse clinical outcomes, including all-cause mortality,hospital admissions because of heart failure, myocardial infarction, and stroke in patients with newly diagnosed heart failure.
Abstract:
- Objective: The authors may need to explicitly state in the objective the proposed benefit from evaluating and displaying the rates of the adverse clinical outcomes in patients with newly diagnosed heart failure. What is the gain the clinical field would incur from this evaluation?
- Results: Please spell out the abbreviation “NYHA” in the text.
Introduction:
- Page 1, line 39: “…and admission for HF”. The reviewer suggests replacing with “…and frequent hospital admissions”.
- Page 1, line 40: “…will die”. The reviewer suggests replacing with “…are expected to die”.
- Page 1, lines 41-44: “In addition, HF hospitalizations are a common complication …….., but also during the first months after the index event”. The sentence is confusing. The reviewer suggests rephrasing.
- Page 2, line 46: “…other adverse outcomes”. Please specify with reference, where appropriate.
- Page 2, lines 53-56: “However, only a few number of HF clinical trials have analyzed the effects of active treatments on other endpoints… infarction or stroke.”. Please rephrase.
- Page 2, lines 66-67: “…the factors associated with the development of HF hospitalization or…”. The reviewer suggests replacing “the development” with “increasing the risk”.
Methods:
- Page 2, lines 95-96: “HF hospitalizations were determined as an inpatient (primary or any diagnosis) with andICD-10 code for any HF”. Please delete “and”.
- Page 2, lines 86-87: As to classification of HF based on LVEF; Could the authors clarify why the patients with unknown/unspecified EF (HFuEF) have been considered a group to compare with. Since the EF is unknown, could not those patients have preserved or reduced EF? Since the evaluation was based on EF phenotype, this group should have been excluded based on lack of echocardiographic data. Please clarify.
- The authors divided the HFpEF into two subtypes (EF 50-60 and EF >60) and presented the associated data of the 2 subtype groups accordingly in the tables; however, the comparisons throughout the manuscript did not actually differentiate between them. Were there included in the intergroup statistical analysis? what are the clinical significance of this classification? Please clarify.
Statistical analysis:
- Page 3, lines 103-104: “Incidence rates were calculated in the overall cohort of HF patients and according to EF phenotype”. What “Incidence rates” the authors refer to? HF incidence or incidence rate of clinical outcomes? Please clarify and edit/delete accordingly.
Results:
- Pages 3-4, lines 147-158: the authors presented the differences between HF phenotypes in clinical profile and HF management. Could the authors clarify which statistical analysis(es) was/were performed and refer it/them in “statistical analysis” section.
- Importantly, the authors should mention the P values of those comparisons with statistically significant differences. Moreover, P values should be also included in table(s) with corresponding statistical analyses mentioned in the footnote of these table(s).
- Page 4, lines 153-156: “Similarly, the prescription of disease modifying HF drugs”. Could the authors specify in the text what are the “disease-modifying HF drugs”?
Tables and Figures:
- All Tables: Please make sure to indicate for ALL continuous variables by “mean and SD”. [Example: Table 1, “Age” should be indicated by “mean ± SD”] and ALL categorical variables by “n (%)” and list the corresponding expression beside each parameter [Example table 2: all parameters are indicated by (%) instead of “n (%)”]. Please edit accordingly.
- All tables and figures: Please make sure to list ALL abbreviations in the table footnote or figure legend, where appropriate. Example “ARNI” in table 1.
- Table 3: “HF hospitalization and mortality”. Do the authors mean death after admission? If yes, please rephrase accordingly.
- Supplementary Figure 1: “No at risk” and “Cum no of events”. Please spell out correctly.
Discussion:
- Page 7, lines 330-333: “However, as the incidence of HF shows a steady increase, together with the ageing of the population and the better treatment of acute cardiovascular events, the number of patients with HF are predicted to increase in the following years”. The sentence may be confusing for the reader, how can the better treatment be a factor for increasing the number of HF patients. Please rephrase, if possible.
- Page 7, line 365: “dapagliflozin”. Can the authors add the type/class of this medication.
- Page 7, line 371: “…the increase after one year follow-up …”. Can the authors clarify what increase they are referring to? Is it increased number of patients taking each medication?
- Page 7, line 386: “…adverse events…”. Please clarify/specify?
- Page 7, line 400: Please spell out the abbreviation “AF”?
Minor:
- Title: Please remove the period (.) at the end of the manuscript title.
- Author affiliations: Please refer to the corresponding author by (*).
- Please separate the “Study Limitations” and Conclusion(s) from Discussion, into 2 different titles.
- Figure 1: It is advisable to plot the 6 panels of this figure on only 1 page, if journal guidelines allow.
Author Response
First, we would like to sincerely thank both reviewers for their comments, as they have clearly been very helpful to improve the quality of the manuscript.
Reviewer 1
jcm-2209374, Burden of Illness Beyond Mortality and Heart Failure Hospitalizations in Newly Diagnosed Heart Failure Patients in Spain According to Ejection Fraction by Carlos Escobar et al. The authors aimed in their study to describe patient characteristics, rates of adverse clinical outcomes, including all-cause mortality, hospital admissions because of heart failure, myocardial infarction, and stroke in patients with newly diagnosed heart failure.
Abstract:
- Objective: The authors may need to explicitly state in the objective the proposed benefit from evaluating and displaying the rates of the adverse clinical outcomes in patients with newly diagnosed heart failure. What is the gain the clinical field would incur from this evaluation?
The abstract has been modified, as suggested.
- Results: Please spell out the abbreviation “NYHA” in the text.
NYHA has been spelled out.
Introduction:
- Page 1, line 39: “…and admission for HF”. The reviewer suggests replacing with “…and frequent hospital admissions”.
It has been changed.
- Page 1, line 40: “…will die”. The reviewer suggests replacing with “…are expected to die”.
It has been changed.
- Page 1, lines 41-44: “In addition, HF hospitalizations are a common complication …….., but also during the first months after the index event”. The sentence is confusing. The reviewer suggests rephrasing.
The sentence has been rephrased.
- Page 2, line 46: “…other adverse outcomes”. Please specify with reference, where appropriate.
The reference has been included.
- Page 2, lines 53-56: “However, only a few number of HF clinical trials have analyzed the effects of active treatments on other endpoints… infarction or stroke.”. Please rephrase.
The sentence has been rephased.
- Page 2, lines 66-67: “…the factors associated with the development of HF hospitalization or…”. The reviewer suggests replacing “the development” with “increasing the risk”.
It has been replaced.
Methods:
- Page 2, lines 95-96: “HF hospitalizations were determined as an inpatient (primary or any diagnosis) with andICD-10 code for any HF”. Please delete “and”.
It has been deleted.
- Page 2, lines 86-87: As to classification of HF based on LVEF; Could the authors clarify why the patients with unknown/unspecified EF (HFuEF) have been considered a group to compare with. Since the EF is unknown, could not those patients have preserved or reduced EF? Since the evaluation was based on EF phenotype, this group should have been excluded based on lack of echocardiographic data. Please clarify.
This subgroup included those patients in which the EF was not collected in the clinical history. As a result, as this was a retrospective study, EF could not be determined. In clinical practice there is a proportion of patients for whom a recent EF is not available (e.g. elderly population, patients with a previous diagnosis of HF, etc.). In order not to introduce a bias in the analysis of the data, we preferred to maintain this group. In our study, this group can include patients with any EF, therefore, HFrEF, HFmrEF and HFpEF. But as it was impossible to differentiate between these patients, we considered that it was preferable to include them as a separate group, defined as unknown EF. We have replaced unspecified EF by unknown EF, in order to clarify this point.
- The authors divided the HFpEF into two subtypes (EF 50-60 and EF >60) and presented the associated data of the 2 subtype groups accordingly in the tables; however, the comparisons throughout the manuscript did not actually differentiate between them. Were there included in the intergroup statistical analysis? what are the clinical significance of this classification? Please clarify.
We divided HFpEF patients into two subgroups, with the aim of determining whether there were any differences between these patients in terms of clinical profile, management or outcomes according to EF. The reason for this was the results from PARAGON-HF (N Engl J Med 2019; 381:1609-1620) and EMPEROR-Preserved (N Engl J Med 2021; 385:1451-1461) studies which showed that the efficacy of sacubitril-valsartan and empagliflozin, respectively, was different in these two subgroups of HFpEF.
Statistical analysis:
- Page 3, lines 103-104: “Incidence rates were calculated in the overall cohort of HF patients and according to EF phenotype”. What “Incidence rates” the authors refer to? HF incidence or incidence rate of clinical outcomes? Please clarify and edit/delete accordingly.
This has been clarified.
Results:
- Pages 3-4, lines 147-158: the authors presented the differences between HF phenotypes in clinical profile and HF management. Could the authors clarify which statistical analysis(es) was/were performed and refer it/them in “statistical analysis” section.
The statistical analysis section has been expanded.
- Importantly, the authors should mention the P values of those comparisons with statistically significant differences. Moreover, P values should be also included in table(s) with corresponding statistical analyses mentioned in the footnote of these table(s).
P values have been included in the tables.
- Page 4, lines 153-156: “Similarly, the prescription of disease modifying HF drugs”. Could the authors specify in the text what are the “disease-modifying HF drugs”?
It has been specified in the text.
Tables and Figures:
- All Tables: Please make sure to indicate for ALL continuous variables by “mean and SD”. [Example: Table 1, “Age” should be indicated by “mean ± SD”] and ALL categorical variables by “n (%)” and list the corresponding expression beside each parameter [Example table 2: all parameters are indicated by (%) instead of “n (%)”]. Please edit accordingly.
The tables have been edited accordingly.
- All tables and figures: Please make sure to list ALL abbreviations in the table footnote or figure legend, where appropriate. Example “ARNI” in table 1.
Abbreviations have been checked.
- Table 3: “HF hospitalization and mortality”. Do the authors mean death after admission? If yes, please rephrase accordingly.
It refers after discharge. It has been clarified.
- Supplementary Figure 1: “No at risk” and “Cum no of events”. Please spell out correctly.
It has been corrected.
Discussion:
- Page 7, lines 330-333: “However, as the incidence of HF shows a steady increase, together with the ageing of the population and the better treatment of acute cardiovascular events, the number of patients with HF are predicted to increase in the following years”. The sentence may be confusing for the reader, how can the better treatment be a factor for increasing the number of HF patients. Please rephrase, if possible.
The sentence has been rephrased.
- Page 7, line 365: “dapagliflozin”. Can the authors add the type/class of this medication.
It has been added.
- Page 7, line 371: “…the increase after one year follow-up …”. Can the authors clarify what increase they are referring to? Is it increased number of patients taking each medication?
It has been clarified in the text.
- Page 7, line 386: “…adverse events…”. Please clarify/specify?
It has been clarified in the text.
- Page 7, line 400: Please spell out the abbreviation “AF”?
The abbreviation AF has been spelled out.
Minor:
- Title: Please remove the period (.) at the end of the manuscript title.
It has been removed.
- Author affiliations: Please refer to the corresponding author by (*).
It has been added.
- Please separate the “Study Limitations” and Conclusion(s) from Discussion, into 2 different titles.
They have been separated.
- Figure 1: It is advisable to plot the 6 panels of this figure on only 1 page, if journal guidelines allow.
We modified the figures.
Reviewer 2 Report
The authors did a good job demonstrating the burden of heart failure in Spain regarding adverse clinical outcomes according to ejection fraction subgroups. This report describes not only hospitalization and mortality but also other major adverse cardiac events. Although data is limited to Spanish patients, it included around 20 thousand patients. The results are not surprising but could be more interesting if more information could be included, my suggestions for improvement are listed below:
1. The pharmacotherapy was included in the study only as a descriptive summary. I do understand that there are only around 10% patients are on SGLT2i, based on the recent clinical trials on this particular drug, it will be interesting to know if SGLT2i’s utilization reduced MACE, hospitalization and all-cause mortality in this small group of patients compared with those were not on SGLT2i.
2. For figure 2, P value should be provided.
3. A major concern about figure 2: it is about the predictors of the risk of the composite outcome hospitalization for HF and all-cause mortality in the overall HF cohort. Please define the composite outcome and explain how these was combined in the analysis.
4. The unit was not uniform throughout the manuscript: per 100 person-years was used to report the HF incidence but per 1000 person-years was used in other places.
Author Response
First, we would like to sincerely thank both reviewers for their comments, as they have clearly been very helpful to improve the quality of the manuscript.
Reviewer 2.
The authors did a good job demonstrating the burden of heart failure in Spain regarding adverse clinical outcomes according to ejection fraction subgroups. This report describes not only hospitalization and mortality but also other major adverse cardiac events. Although data is limited to Spanish patients, it included around 20 thousand patients. The results are not surprising but could be more interesting if more information could be included, my suggestions for improvement are listed below:
- The pharmacotherapy was included in the study only as a descriptive summary. I do understand that there are only around 10% patients are on SGLT2i, based on the recent clinical trials on this particular drug, it will be interesting to know if SGLT2i’s utilization reduced MACE, hospitalization and all-cause mortality in this small group of patients compared with those were not on SGLT2i.
In our study the proportion of patients taking SGLT2i was very low, as in Spain the SGLT2i approval for their use in HF patients was at the end of 2021. This makes that the impact of SGLT2i on adverse events cannot be properly analyzed in our study. However, at this moment we are analyzing this point in a separate specific study. It is expected that in real life this would be the case based on studies performed with SGT2i, such as empagliflozin (EMPEROR-Reduced, EMPEROR-Preserved) and dapagliflozin (DAPA-HF, DELIVER) in all spectrum of heart failure ejection fractions.
- For figure 2, P value should be provided.
The P values of figure 2 have been included.
- A major concern about figure 2: it is about the predictors of the risk of the composite outcome hospitalization for HF and all-cause mortality in the overall HF cohort. Please define the composite outcome and explain how these was combined in the analysis.
To perform this figure we analyzed time to first event, either HF hospitalization or all-cause death. This has been clarified in the figure.
- The unit was not uniform throughout the manuscript: per 100 person-years was used to report the HF incidence but per 1000 person-years was used in other places.
We have homogenized the results throughout the manuscript.
Round 2
Reviewer 1 Report
The reviewer thanks the authors for incorporating the suggested changes. However, some comments still need to be further addressed. Please see below:
1- Page 2, line 46: The authors cited the requested reference; however, they still need to further explain what those adverse outcomes in the text. Please edit the text accordingly.
2- In reply to the question why HFpEF group was divided into two subtypes (EF 50-60 and EF >60), the authors claimed that they aimed to determine whether there were any differences between these patients in terms of clinical profile, management, or outcomes according to EF. Still, the authors did NOT present any data discussing any statistical analyses have been done for comparing these 2 subgroups. Please clarify and edit the text accordingly with the appropriate statistical analysis, whenever possible.
3- Statistical analysis:
Page 3, lines 110-114: In response to the first round of review regarding statistical analyses used, the authors added the following sentence “To analyze the association between continuous variables amongst EF subtypes, the two-sample t-test was used for variables normally distributed and the Mann Whitney U test for those non-normally distributed. The chi-square test was used for categorical variables”.
3.a. “the association”. Please use alternative expression given that the t-test is used to compare the means of the groups not the association (correlation).
3.b. Please specify which set of data (not-normally distributed) that was analyzed with Mann Whitney U test?
3.c. The authors used t-test to compare the HF groups (HFmrEF vs. HFrEF & HFpEF vs. HFrEF); however, they did not compare HFpEF vs. HFmrEF. Please clarify and edit accordingly.
3.d. For comparing more than 2 groups (like the case here), one-way ANOVA is considered more powerful than the t-test as it provides more comprehensive analysis of the data while controlling for the risk of Type I error. Accordingly, could the authors clarify why they opted to choose to perform multiple t-test over the one-way ANOVA?
4- Results: The authors need to be more specific when comparing groups. For example: Page 4, lines 153-157 : “Patients with HFpEF were older (p =0.001), more predominantly female (p <0.001), and had higher prevalence of atrial fibrillation (p <0.001), at baseline. Patients with HFrEF had higher prevalence of type 2 diabetes (p <0.001), chronic kidney disease (p <0.001), coronary artery disease (p <0.001), stroke (p <0.001) and peripheral artery disease (p <0.001), at baseline“. The authors claim here that Patients with HFpEF were older, more predominantly female, with higher prevalence of AF; however, they did not specify which group they are comparing with, especially that according to table 1, it seems that HFpEF only compared with HFrEF but NOT against HFmrEF? Please specify/clarify and apply to the whole “Results” section, where appropriate.
5- Tables and figures:
5.a. Please specify the statistical analysis in the footnote of ALL tables.
5.b. Table 1: Please use the format “mean ± SD” for continuous parameters. Example: Age should be “69.7 ± 19” NOT “69.7 (19). The latter format is used preferably with categorical variables n (%). Please edit accordingly.
5.c. Supplementary Figure 1: “No at risk” Please spell out as “Number at risk” or “No. at risk”.
Author Response
1- Page 2, line 46: The authors cited the requested reference; however, they still need to further explain what those adverse outcomes in the text. Please edit the text accordingly.
The sentence has been completed.
2- In reply to the question why HFpEF group was divided into two subtypes (EF 50-60 and EF >60), the authors claimed that they aimed to determine whether there were any differences between these patients in terms of clinical profile, management, or outcomes according to EF. Still, the authors did NOT present any data discussing any statistical analyses have been done for comparing these 2 subgroups. Please clarify and edit the text accordingly with the appropriate statistical analysis, whenever possible.
We have completed tables 1-3, comparing these 2 subgroups and the discussion has been expanded accordingly.
3- Statistical analysis:
Page 3, lines 110-114: In response to the first round of review regarding statistical analyses used, the authors added the following sentence “To analyze the association between continuous variables amongst EF subtypes, the two-sample t-test was used for variables normally distributed and the Mann Whitney U test for those non-normally distributed. The chi-square test was used for categorical variables”.
This section has been updated.
3.a. “the association”. Please use alternative expression given that the t-test is used to compare the means of the groups not the association (correlation)..
It has been changed
3.b. Please specify which set of data (not-normally distributed) that was analyzed with Mann Whitney U test?
The only continuous variable that was compared was age, that was normally distributed. So, this test has not been used, but we have still kept it in the statistical analysis method section.
3.c. The authors used t-test to compare the HF groups (HFmrEF vs. HFrEF & HFpEF vs. HFrEF); however, they did not compare HFpEF vs. HFmrEF. Please clarify and edit accordingly.
As the prevalence of comorbidities in patients with HFmrEF were inter-mediate between HFrEF and HFpEF, we aimed to compare patients HFpEF vs HFrEF. This has been clarified in the text.
3.d. For comparing more than 2 groups (like the case here), one-way ANOVA is considered more powerful than the t-test as it provides more comprehensive analysis of the data while controlling for the risk of Type I error. Accordingly, could the authors clarify why they opted to choose to perform multiple t-test over the one-way ANOVA?
The reason to use the two-sample t-test was that the HFrEF group is the reference group for the comparison with other HF phenotypes. In addition, the ANOVA approach would be more powerful to detect an overall "effect", however, it was not the goal of the analysis in this study.
4- Results: The authors need to be more specific when comparing groups. For example: Page 4, lines 153-157 : “Patients with HFpEF were older (p =0.001), more predominantly female (p <0.001), and had higher prevalence of atrial fibrillation (p <0.001), at baseline. Patients with HFrEF had higher prevalence of type 2 diabetes (p <0.001), chronic kidney disease (p <0.001), coronary artery disease (p <0.001), stroke (p <0.001) and peripheral artery disease (p <0.001), at baseline“. The authors claim here that Patients with HFpEF were older, more predominantly female, with higher prevalence of AF; however, they did not specify which group they are comparing with, especially that according to table 1, it seems that HFpEF only compared with HFrEF but NOT against HFmrEF? Please specify/clarify and apply to the whole “Results” section, where appropriate.
This paragraph has been expanded in order to be more informative.
5- Tables and figures:
5.a. Please specify the statistical analysis in the footnote of ALL tables.
It has been specified.
5.b. Table 1: Please use the format “mean ± SD” for continuous parameters. Example: Age should be “69.7 ± 19” NOT “69.7 (19). The latter format is used preferably with categorical variables n (%). Please edit accordingly.
It has been changed.
5.c. Supplementary Figure 1: “No at risk” Please spell out as “Number at risk” or “No. at risk”.
It has been modified.